# Supramolecular Atropine Potentiometric Sensor

**DOI:** 10.3390/s21175879

**Published:** 2021-08-31

**Authors:** Catarina Ferreira, Andreia Palmeira, Emília Sousa, Célia G. Amorim, Alberto Nova Araújo, Maria Conceição Montenegro

**Affiliations:** 1LAQV/REQUIMTE, Departamento de Ciências Químicas, Faculdade de Farmácia, Universidade do Porto, R. Jorge Viterbo Ferreira 228, 4050-313 Porto, Portugal; up201305083@ff.up.pt (C.F.); anaraujo@ff.up.pt (A.N.A.); mcbranco@ff.up.pt (M.C.M.); 2Laboratório de Química Orgânica e Farmacêutica, Faculdade de Farmácia, Universidade do Porto, Rua de Jorge Viterbo Ferreira 228, 4050-313 Porto, Portugal; apalmeira@ff.up.pt (A.P.); esousa@ff.up.pt (E.S.); 3CIIMAR—Centro Interdisciplinar de Investigação Marinha e Ambiental, Terminal de Cruzeiros do Porto de Leixões, 4450-208 Matosinhos, Portugal

**Keywords:** potentiometry, ion-selective electrodes, cucurbit[6]uril, atropine, pharmaceutical formulations

## Abstract

A supramolecular atropine sensor was developed, using cucurbit[6]uril as the recognition element. The solid-contact electrode is based on a polymeric membrane incorporating cucurbit[6]uril (CB[6]) as an ionophore, 2-nitrophenyl octyl ether as a solvent mediator, and potassium tetrakis (4-chlorophenyl) borate as an additive. In a MES-NaOH buffer at pH 6, the performance of the atropine sensor is characterized by a slope of (58.7 ± 0.6) mV/dec with a practical detection limit of (6.30 ± 1.62) × 10^−7^ mol/L and a lower limit of the linear range of (1.52 ± 0.64) × 10^−6^ mol/L. Selectivity coefficients were determined for different ions and excipients. The obtained results were bolstered by the docking and spectroscopic studies which demonstrated the interaction between atropine and CB[6]. The accuracy of the potentiometric analysis of atropine content in certified reference material was evaluated by the *t*-Student test. The herein proposed sensor answers the need for reliable methods providing better management of this hospital drug shelf-life while reducing its flush and remediation costs.

## 1. Introduction

The reduction in the impact of pharmaceutical substances on the environment is a current topic included in the worldwide pharmaceutical strategy to contribute to climate neutrality. Managing already-open-hospital drugs can decrease the impact of these substances in the environment with potential cost benefits to the hospital. The development of ion-selective electrodes based on specific recognition elements for a target drug substance, in a quality control context, can be the way to reach those goals.

Atropine is a natural amine extracted from leaves of the deadly nightshade (Atropa belladonna) and owes its name to the inflexible Atropos from Greek mythology, one of the three goddesses assigning destinies to mortals at birth. Descriptions of its use date back to before Christ and range from dilation of pupils, bringing allure to the look of lovers up to the treatment of wounds, gout, and sleeplessness. This alkaloid drug, commonly used in prehospital and emergency departments, has nowadays important applications as an ophthalmic agent, because of its cycloplegic and mydriatic action, in resuscitation after cardiac dysrhythmia and heart block, and as an antidote in organophosphate poisoning because of its antagonist effect over the muscarinic acetylcholine [1]. Atropine has also been associated with a deadly poison [2], making its determination very important in many different aspects.

Different analytical methods are reported in the literature for its determination, such as spectrophotometry [3], electrochemistry [4], chemiluminescence [5,6], gas chromatography, and high-performance liquid chromatography. However, these methods require more laborious manipulation, expensive reagents, and sophisticated instruments. Meanwhile, potentiometric methods based on the use of ion-selective electrodes appeared as an alternative because of their inherent advantages over those methods, such as portability and real-time analysis [7,8,9,10,11]. The use of ion PVC sensors, where the plasticizer was dopped with sparingly soluble atropine salts, such as atropine-reineckate [8], atropine-phosphotungstate [10], atropine-tetrakis(4-chlorophenyl)-borate [9], or simply potassium tetrakis-[3,5-bis-(trifloromethyl)-phenyl] borate [8,9,10], enabled simplified potentiometric determination based on the exchange equilibrium with the sample solution, though with low selectivity trade-off [12]. In turn, the use of neutral ionophores, such as β-cyclodextrin [10], phosphorated calix[6]arene derivatives [13] or valinomycin [14], looks to have better sensor characteristics concerning the selectivity as well as a larger linear response.

Cucurbit[n]uril family are supramolecular host molecules made of glycoluril units bridged by methylene groups obtained after condensation reactions [15]. The trivial name derives from structure resemblance to a pumpkin (botanical family Cucurbitaceae) and are named according to the number n of glycoluril units [16]. Different literature reviews assigned to cucurbiturils (CBs), their homologues, and adducts, have provided new opportunities in many areas in supramolecular chemistry including separation, transport, recognition, catalysis, and sensors due to their rigid structure, selectivity, and the capacity of forming stable inclusion complexes with molecules and ions [17,18]. CB[7] was already studied as a drug delivery system concerning the improvement of drug bioavailability, increase targeting, and diminishing a drug’s systemic toxicity. CB[7] showed an enhanced availability of atropine in the central nervous system [19]. 

Based on the aspects previously reported, this work focusses on the reliable determination of atropine by employing potentiometric sensors dopped with cucurbituril CB[6]. The performance as well the response mechanism of the sensor is further interpreted with the addition of docking studies and spectroscopic techniques, such as IR and NMR.

## 2. Materials and Methods

### 2.1. Materials

#### 2.1.1. Reagents and Solutions

Analytical grade chemicals were used without further purification unless otherwise stated. Atropine sulfate (K1570875) was purchased from Merck^®^ (Darmstadt, Germany); poly(vinyl chloride) carboxylated (PVC-COOH) (18.311.95) was purchased from Janssen Chimica^®^ (Beerse, Belgium); cucurbit[6]uril hydrate (CB[6]) (94544-1G-F), tetrahydrofuran (THF) (186522-2L), 2-fluorophenyl 2-nitrodiphenyl ether (2-FNDPE) (4790-5ML-F), calcium chloride (C8106-500G), dibutyl sebacate (DBS) (84838-5ML), MES hydrate (M8250-25G), Trizma® hydrochloride (T3253), polysorbate 80 (59924), disodium EDTA (ED2SS), lithium chloride (310468-500G), sodium citrate dihydrate (S1804-500G), sodium phosphate (342483), and benzalkonium chloride (12060) were purchased from Sigma-Aldrich^®^ (St. Louis, MO, USA); 2-nitrophenyl octyl ether (2-NPOE) (73732-25ML), potassium tetrakis(4-chlorophenyl)borate (KTpCIPB) (60591), ammonium chloride (09702), tetrapentylammonium bromide (TPAB) (88001), and boric acid (15660) were purchased form Fluka^®^ (Buchs, Switzerland); benzylic alcohol (100-51-6) was purchased from José M. Vaz Pereira^®^ (Lisboa Portugal); sodium chloride (7647-14-5) was purchased from José Manuel Gomes dos Santos, Lda^®^ (Odivelas, Portugal); dibasic sodium phosphate (30412) was purchased from Riedel de Haën^®^ (Seelze, Niedersachsen, Germany); and potassium chloride (7447-40-7) and sodium hydroxide (1310-73-2) were purchased from AnalaR NORMAPUR^®^ (Radnor, PA, USA).

All aqueous solutions were prepared with doubly deionized Milli-Q water (Heal force; Shanghai; China) (conductivity < 0.1 μS/cm). Atropine stock solutions were prepared daily by weighing about 35 mg of reagent into a 50-mL volumetric flask followed by dilution to the mark with a 0.01-mol/L calcium chloride solution acting as an ionic strength adjuster (I = 0.03 mol/L) or with a 0.01-mol/L MES buffer solution. The calibrating working solutions were prepared from the stock by further dilution.

#### 2.1.2. Apparatus

A Crison 2002 micro digital meter (sensitivity ± 0.1 mV) coupled to an Orion 605 electrode switcher from Thermo Fisher Scientific (Waltham, MA, USA) was used to measure the potential differences between the atropine electrodes and the reference electrode at 25 °C. The last consisted of a silver chloride/silver double junction electrode (Orion 90-02-00), with the external compartment filled with a 0.01 mol/L CaCl_2_ solution. The pH measurements were performed with a Crison pH electrode coupled to a pH Meter GLP22—Crison (Barcelona, Spain).

A Fourier transform infrared (FTIR) spectrometer from PerkinElmer Frontier (Beaconsfield, UK) equipped with an attenuated total reflectance (ATR) accessory with a pressure arm to control the applied force and reduce sample-to-sample variability was used in the study of the interaction between atropine and CB[6]. Baseline correction, normalization, and peak positions were determined for all spectra by Spectrum software v.5.3.1., from the same brand.

^1^H NMR spectra were taken in DMSO-d6 at room temperature, on Bruker Avance 300 instrument (300.13 MHz; Wissembourg, France).

### 2.2. Methods

#### 2.2.1. Membrane Preparation and Electrode Construction

Six types of electrodes differing on the composition of the selective membrane, as stated in Table 1, were prepared. Each mixture of the ionophore, plasticizer, and ionic additive was further mixed with the polymeric matrix and carboxylated polyvinylchloride, previously dissolved in THF (6 mL). The membrane solution was then dropped directly on the conductive surface of the electrode and left to dry for 24 h. The conductive surface was made up of a mixture of epoxy resin (Araldite M) with graphite powder following the procedure already described [20]. Before evaluation, the electrodes were soaked in deionized water for at least 30 min to promote membrane hydration.

#### 2.2.2. Electrode Characterization

The evaluation of the atropine electrodes was firstly performed by three successive calibrations using simultaneously three different electrodes bodies with the studied mem-brane. The atropine standard solutions were added in the concentration range 9.0 × 10^−7^ up to 1.0 × 10^−2^ mol/L and vice-versa, with the ionic strength adjusted to 0.01 mol/L by the addition of CaCl_2_ salt. The potential readings were registered after stabilization (±0.2 mV). After each calibration, the electrodes were carefully washed in water for at least 30 min. After that, they were reevaluated under the described conditions. The practical detection limit (PDL) was taken from the calibration plot as the abscissa of the intersection point of the extrapolated linear segments, corresponding, respectively, to the absence of response for lower concentrations and the concentrations interval translated by Equation (1). In this last interval, the lower concentration was the so-called lower limit of linear response (LLLR). The effect of pH on the electrode potential change was evaluated for two atropine solutions (10^−4^ and 10^−5^ mol/L), in the pH range of 2–11, by the small volume additions of concentrated H_2_SO_4_ or NaOH. The potentiometric selectivity coefficients for the most common anions presented in the sample matrix were assessed through the separated solutions method [20]. Therefore, the potential difference of two separate solutions with the same activity, one containing the atropine ion (A) and the other containing the interferent ion (I), was measured and the corresponding coefficient was calculated according to Equation (1).
(1)logKA,IPot=E2−E12.303RT/zAF+(1−zAzi)logaA
where $a_{A}$ is the activity of the primary ion; $z_{A}$ and $z_{I}$ are the charges of the primary and interfering ion, respectively; $E_{2}$ and $E_{1}$ are the measured potential at the same activity of the primary ion and the interfering ion, respectively; *R* is the universal gas constant; *F* is the Faraday constant; and T is the absolute temperature.

The determination of atropine concentration in certified reference material by the proposed method was made up by a direct dilution in 0.01 mol/L MES-NaOH buffer solution, pH 6.

#### 2.2.3. Docking Studies

The three-dimensional structure of CB[6] necessary for the docking study of atropine was obtained from Cambridge Crystallographic Data Centre (CCDC) (Deposition number 1540086) [21]. Structures of test molecule atropine and control molecules ephedrine [22], isoprenaline [23], octopamine [22], synephrine [22], lidocaine [24], prilocaine, and procaine [24] were obtained from Pubchem [25] and minimized by the semiempirical Polak–Ribiere conjugate gradient method (RMS < 0.1 kcal/Å/mol) [26] using HyperChem 7.5 (Hypercube, Gainesville, FL, USA) [27]. Structure-based docking was carried out using AutoDock Vina (Molecular Graphics Lab, San Diego, CA, USA) [28]. A grid box covering the entire CB[6] structure was built, and default settings for small molecule-protein docking were used throughout the simulations. The top 9 poses were collected for each molecule and the lowest docking score value was associated with the more favorable binding conformation. PyMol1.3 (Schrödinger, New York, NY, USA) [29,30] was used for visual inspection of results.

#### 2.2.4. Spectroscopic Analysis of the Complex of Atropine and CB[6]

IR spectra were obtained by mixing accurately weighed 0.3 mg of atropine and 15 mg of CB[6] with further kneading in an agate mortar for 10 min [31]. Briefly, few drops of water were added to obtain a homogeneous paste. The resulting paste was dried in an oven at 45 °C for 24 h. The solid obtained was pulverized before analysis. Vibrational spectra with 8 cm^−1^ resolution were collected in the wavenumber range of 4000–600 cm^−1^ (32 scans). The background was made with the ATR accessory empty.

## 3. Results

### 3.1. Evaluation of the Electrode Behaviour

The solvent mediator was the first constituent under optimization to attain the atropine-selective electrode with optimal performance. It determines the viscosity of the membrane and the mobility of ions/molecules within that phase, but mainly its lipophilicity [32] and the membrane selectivity as a result. Three membrane compositions (type I, II and III) were prepared using solvent mediators with different increasing lipophilicity (FNDPE (XlogP3 = 3.4), oNPOE (XlogP3 = 5.1), and DBS (XlogP3 = 5.3)) [33]. For these membranes, Nernstian responses were obtained, congruent with the positive single charged atropine (Table 2). However, for the more lipophilic solvent mediator DBS, a slight decrease of about 8% in the slope, S, was noticed. On the contrary, 2-NPOE provided the highest calibration slope together with the improvement of the linear response range, LLLR, in almost half a concentration decade. 

The electrodes prepared with membranes II and III which exhibited a larger linear response range were selected to evaluate the effect of increasing or decreasing the number of negatively charged sites already introduced by the carboxyl functionalities of the PVC polymer. Thus, the absence of lipophilic salt (KTpCIPB), which led to the addition of negative sites to the membrane as well as a replacement for the TPAB salt, bringing positive sites, was considered in new electrodes prepared with the membranes type IV to VI (Table 1). The borate salt elimination from membranes formulation caused a more negative effect on the membrane-based on DBS (Type V) than on the membrane based on 2-NPOE (Type IV), concerning the slope, PDL, and LLLR, while lowering the readings’ reproducibility. The replacement of this negative lipophilic salt for a positive, TPAB (type VI), blocked the atropine interaction with the membrane, not being possible to observe any variation of the potential with the logarithm of the atropine activity. The presence of lipophilic anion (KTpCIPB) improved the ion extraction and ensured the perm-selective of the sensing membrane, explaining its importance in the membrane composition. The type II membrane was selected for further studies, once the main electrode characteristics such as LLLR were much more competitive than other electrodes reported by Alçada et al. [9] (1.2 × 10^−5^ mol/L), Mostafa et al. [10] (1 × 10^−6^ mol/L), Zareh et al. [13,14] (1.9 × 10^−6^ mol/L) or even by using an electrochemiluminescent-based sensor [34].

The effect of pH in the potential of the electrodes was also evaluated for two atropine solutions (1.00 × 10^−5^ mol/L and 1.00 × 10^−4^ mol/L). A negative correlation between the potential and the pH was observed. These results were expected because of the formation of the non-ionized form of the atropine above its pKa (9.43). As the potential was strongly dependent on pH, the potentiometric response was determined in different buffer solutions. According to Table 3, the main electrode characteristics were improved until the pH reached 6. By comparing MES-NaOH (pH = 6.0) with Tris.HCl-NaOH (pH = 6.5), a big decay was noticed in the sensor characteristics that imputed not only to the small pH variation but mainly to lower selectivity of the electrode to the molecules that were used to prepare TRIS-HCl or CH3COOH buffer. So, to adjust the pH during atropine calibrations, an MES-NaOH (pH = 6) buffer was chosen to ensure a total atropine ionization, avoiding the presence of more interferent species.

As selectivity is one of the most important characteristics of an ion-selective electrode, the potentiometric selectivity coefficients $K_{Atropine,Interf}^{Pot}$ were determined according to the separated solutions method to determine the ability of the electrode to selectively respond to the primary ion over other ions present in the solution. The most common ions and molecules present in pharmaceutical atropine formulations were studied at three different concentrations atropine levels (Table 4). As observed, the higher the concentrations of the interferent in the solution under measurement, the lower its potentiometric selectivity coefficient, being the electrode more selective for atropine. Divalent ions are less interferent than monovalent ions. The most common excipients used in formulations present potentiometric selectivity coefficients between 0.01 and 0.6. Benzalkonium chloride is the most interferent species studied here (Table 4).

### 3.2. In Silico Studies of the Atropine—CB[6] Interactions

CB[6] is a typical representative cucurbituril composed of 6 glycouril units linked by methylene bridges and possesses a hydrophobic cavity accessed via two polar carbonyl-rimmed openings [35]. CBs are known to form very stable host–guest inclusion complexes with cationic molecules because of ion–dipole interactions, hydrogen bonding, and hydrophobic interactions [36]. Hence, several crystal structures of CB[6] host–guest complexes are described in the literature [37,38,39], providing the basis to further understand how the atropine molecule interacts as the guest of the cucurbituril. Therefore, binding free energies for known CB[6] guests were predicted by docking and used as positive controls, then compared to the free energy of the CB[6], i.e., the atropine complex. The controls were chosen according to the structural similarity with the test molecule atropine; all test compounds have a methylbenzene group and have the same three features pharmacophore (one aromatic ring, one hydrogen bond donor and one positive ionizable group) (Appendix A). The found free energies ranged from −2.8 kcal/mol for lidocaine down to −3.6 kcal/mol for the more stable isoprenaline or prilocaine, i.e., CB[6] complexes (Table 5). Concerning atropine, the most stable binding conformation exhibited a docking score of −3.4 kcal/mol, which not only places this guest in the same range of binding affinities of the positive controls, but also reveals free energy of binding similar to the ones presented by the top-ranked positive controls isoprenaline and prilocaine.

Because of the volume of atropine, only the hydroxyl group is capable of being lodged in the CB[6] cavity, establishing hydrogen interactions (Figure 1). The three-atoms-long bridge between the azabicycloctane and the benzene ring allows the establishment of polar interactions between those end groups and the CB[6] rims. The molecule presents a hydrophilic character brought by the amine, carbonile, and hydroxyl moieties. The protonable tertiary amino group is suitable for ion–dipole and hydrogen bonding interactions with highly polar carbonyls on the portals of CB[6] (Figure 1). The ester group provides an extra anchoring point for dipole–dipole binding to the host molecules. The hydrophobic portions of atropine are an aromatic ring and a bicyclooctane, which are involved in van der Waals and permanent dipole-induced dipole interactions (Figure 1). Either the hydrophobic effect, as well as ion–dipole, dipole–dipole, and hydrogen interactions, were addressed as the main driving forces for the binding of different guests by CB [36]. In turn, the superficial/partial entrance of the guest molecule into the host was reported for other CB[6] complexes [23,40]. In agreement with the obtained results, several drugs described in the literature also possess protonable amino groups allowing interaction between cationic guests and hosts [24,41]. 

During this study, it was also hypothesized whether the interactions of atropine with CB[6] could change their chemical properties. Solid complexes can be prepared by different methods (kneading, co-evaporation, freeze-drying) [31]. The interactions between the small molecule and the host could be dependent on the technique used. Following our previous experience on the simple kneading method [42], the preparation of the solid complex was performed with energetic kneading [31] and the resulting structure was established by IR and NMR, as depicted in Figure 2. 

Atropine sulfate IR spectrum (Figure 2A—blue spectrum) demonstrates a broadband at 3204 cm^−1^, corresponding to the stretch vibrations of O–H and 1720 cm^−1^ because of in-plane vibrations. CH and CH_3_ stretching and bending bands appeared at 2940 cm^−1^ and 1454 cm^−1^, respectively, and in the fingerprint region, C=C aromatic bonds were present. CB[6] IR spectrum (Figure 2A—orange spectrum) shows broadband at 3470 cm^−1^ and 1720 cm^−1^, corresponding to the vibrations of O–H groups, and bands at 2928 cm^−1^ because of C–H stretching. The stretching frequency of C–N was attributed to the band at 1475 cm^−1^, the C–O bond stretching to the strong band at 1176 cm^−1^, and the rocking vibration of CH_2_ to the band at 802 cm^−1^. Atropine sulfate-CB[6] complex obtained by kneading furnished an IR spectrum (Figure 2A—grey spectrum) with different bands considering both peak intensity and shape. Significant modifications in wavenumber were noted for the bands corresponding to the O–H stretching (3312 cm^−1^) and for the fingerprint region, which led us to hypothesize the establishment of interactions between atropine and CB[6] involving their hydroxyl groups, as previously predicted by docking studies. In the IR spectra of the complex, the 1600 cm^−1^ bands of the benzene ring and the 1100 cm^−1^ adsorption band belonging to the C-N-C moiety of the tropane ring reinforces the inclusion of atropine in CB[6]. Moreover, more diversified types of C-H bond bands were noted at the 2800–2900 cm^−1^ region. 

Atropine-CB[6] mixture was also analyzed by ^1^H NMR spectroscopy (Figure 2B). Both atropine sulfate and CB[6] spectra were similar to those previously described [43,44]. The complex ^1^H NMR spectrum (Figure 2B, grey spectrum) is practically superimposable with the combination of both atropine and CB[6] spectra except for the broad signal at δH 4–5 ppm (Figure 2B, grey area) attributed to the atropine hydroxyl proton. Lower chemical shift values were noted for protons of the CB[6] units, suggesting atropine proximity to these protons.

Noteworthy CB[6] in DMSO-d6 is much more soluble in the presence of atropine than alone. As a result of complex formation, there are physicochemical properties of the guest molecules, such as solubility change. The noted increase in solubility of the complex reinforces the establishment of additional interactions between both chemical entities [31]. Both docking and spectroscopic studies predict the formation of a complex between atropine and CB[6] which comes to support the use of this macromolecule as a suitable substrate for electrode recognition.

### 3.3. Determination of Atropine in Certified Reference Material

The usefulness of the atropine-based sensor was evaluated for its direct determination using a polymeric CB[6] membrane with potassium tetrakis(4-chlorophenyl) borate dissolved in 2-nitrophenyl octyl ether. In this method development application, two sensor units were used, and measurements were made in quadruplicate. The atropine concentrations of (1.79 ± 0.12) × 10^−5^ mol/L and (3.65 ± 0.17) × 10^−5^ mol/L were each measured in the certified sample (1.0 mg mL^−1^ in acetonitrile, ampule of 1 mL, certified reference material, Cerilliant^®^ (Darmstadt, Germany)). These results are under the certified value of the sample (1.72 × 10^−5^ and 3.42 × 10^−5^) mol/L, with acceptance limits between (1.77–1.82) × 10^−5^ mol/L and (3.57–3.74) × 10^−5^ mol/L, since the t values for 95% confidence level is lower than 1.96.

## 4. Conclusions

The developed atropine-PVC membrane sensor described in this work offers an alternative to the more tedious, albeit generic, chromatographic procedures for the determination of atropine in pharmaceutical preparations. A new atropine-selective electrode is proposed, using CB[6] as an ionophore. The incorporation of CB[6], together with a lipophilic anionic additive in membrane composition, enables easy-to-construct sensors with a fast response and good selectivity down to the micromolar level, which are much better than other sensors reported in the literature. These findings were predicted by docking and spectroscopic studies highlighting hydrogen interactions between the atropine hydroxyl group and CB[6]. The atropine selective membrane was successfully applied to certified reference material. The results fit the requirements of the statistical analysis by the *t*-test. The proposed sensor can be helpful in the management of already-open-hospital drugs as a quality control tool.

## Figures and Tables

**Figure 1 sensors-21-05879-f001:**
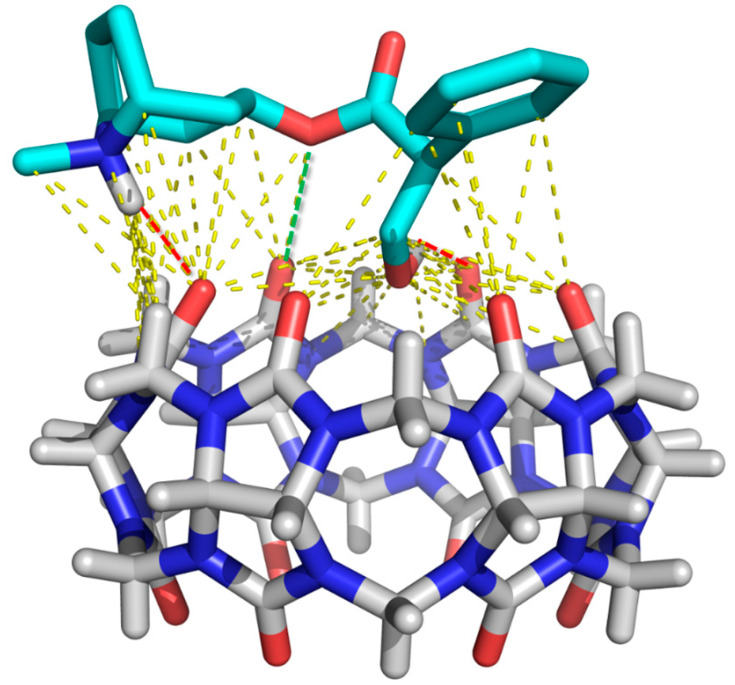
Crystallographic structure of CB[6] (CCDC deposition number 1540086) (white sticks) bound to docked atropine (light blue sticks). Hydrogen interactions and dipole–dipole interactions are represented as red and green broken lines. Other interactions (van der Waals, hydrophobic, dipole-induced dipole) are represented as yellow broken lines. Oxygens and nitrogens are represented in red and blue, respectively.

**Figure 2 sensors-21-05879-f002:**
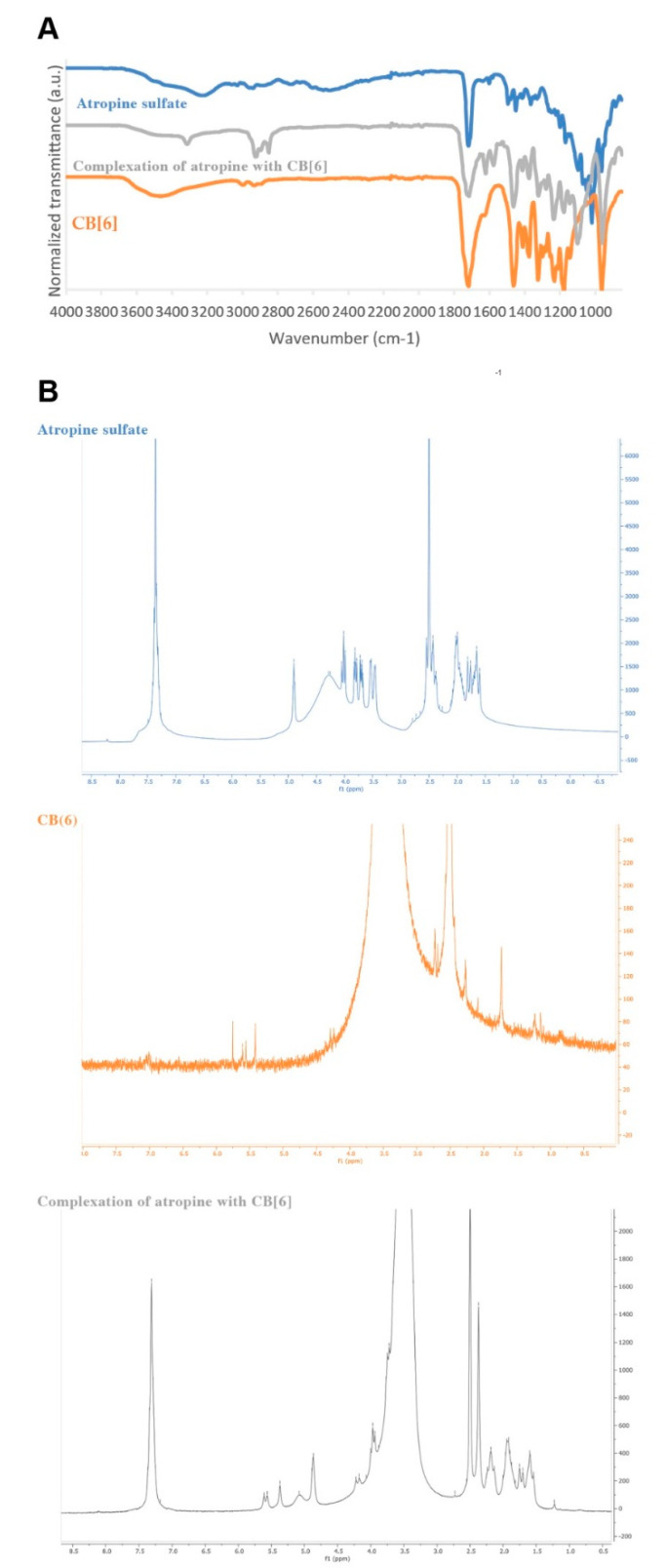
(**A**) IR spectra of atropine, atropine sulfate and CB[6] after kneading, and CB[6] alone (blue, grey, and orange); (**B**) 1H NMR spectrum of atropine sulfate (blue top), CB[6] (orange middle), and atropine and CB[6] after kneading (grey bottom).

**Table 1 sensors-21-05879-t001:** Membrane composition (% *w*/*w*) of the constructed electrodes for atropine.

Type	CB[6]	2-FNDPE	2-NPOE	DBS	KTpCIPB	TPAB	PVC-COOH
	(% *w*/*w*)						
I	1.06	68.62	-	-	0.28	-	30.03
II	1.07	-	68.54	-	0.27	-	30.13
III	0.94	-	-	68.70	0.27	-	30.10
IV	0.90	-	69.88	-	-	-	29.21
V	1.03	-	-	69.83	-	-	29.13
VI	0.98	-	69.01	-	-	0.24	29.77

CB[6], cucurbit[6]uril; FNDPE, 2-fluorophenyl 2-nitrophenyl ether; NPOE, 2-nitrophenyl octyl ether; DBS, dibutyl sebacate; KTpCIPB, potassium tetrakis(4-chorophenyl) borate; TPAB, Tetrapentylammonium bromide PVC-COOH, carboxylated polyvinyl chloride.

**Table 2 sensors-21-05879-t002:** Potentiometric working characteristics for the atropine-selective electrode.

Type	Slope (mV/dec)	LLLR (mol/L)	PDL (mol/L)
I_(n=6)_	59.54 ± 2.18	(6.90 ± 0.24) × 10^−6^	(4.39 ± 1.34) × 10^−7^
II_(n=6)_	60.30 ± 1.07	(1.62 ± 2.27) × 10^−6^	(3.50 ± 1.15) × 10^−7^
III_(n=6)_	55.51 ± 0.42	(2.00 ± 0.00) × 10^−6^	(5.31 ± 1.17) × 10^−7^
IV_(n=6)_	59.84 ± 2.13	(3.80 ± 2.14) × 10^−6^	(4.01 ± 1.75) × 10^−7^
V_(n=6)_	53.85 ± 0.75	(6.33 ± 0.33) × 10^−6^	(2.38 ± 0.63) × 10^−6^

LLLR—Lower limit of linear range; PDL—Practical detection limit.

**Table 3 sensors-21-05879-t003:** Effect of the pH in the potentiometric response.

Buffer Composition	Slope (mV/dec)	LLLR (mol/L)	PLD (mol/L)
HCl-KCl (pH 2.5)_n=4_	(43.66 ± 4.70)	(6.78 ± 0.00) × 10^−5^	(1.99 ± 1.04) × 10^−5^
MES (pH 4.0)_n=3_	(57.70 ± 0.29)	(1.57 ± 1.2) × 10^−6^	(3.04 ± 0.32) × 10^−7^
MES-NaOH (pH = 6.0)_n=11_	(58.72 ± 0.60)	(1.52 ± 0.64) × 10^−6^	(6.30 ± 1.62) × 10^−7^
Tris HCl-NaOH (pH = 6.5)_n=4_	(44.16 ± 0.54)	(9.99 ± 0.00) × 10^−6^	(4.19 ± 0.46) × 10^−6^
CH3COOH-NaOH (pH 6.5)_n=4_	(48.93 ± 0.70)	(1.18 ± 0.00) × 10^−4^	(1.37 ± 0.41) × 10^−5^

**Table 4 sensors-21-05879-t004:** Potentiometric selectivity coefficients $K_{Atropine,Interf}^{Pot}$ for the atropine-selective electrode.

		$K_{Atropine,Interf}^{Pot}$	
	Interfering Species Concentration		
	9.99 × 10^−6^ mol/L	4.76 × 10^−4^ mol/L	5.56 × 10^−3^ mol/L
Calcium chloride	1.42 × 10^−4^	1.90 × 10^−5^	7.61 × 10^−6^
Magnesium chloride	1.92 × 10^−4^	2.05 × 10^−5^	1.06 × 10^−5^
Polysorbate 80	1.23 × 10^−2^	1.14 × 10^−2^	8.63 × 10^−3^
Ammonium chloride	3.50 × 10^−2^	1.02 × 10^−3^	2.48 × 10^−4^
Sodium chloride	3.82 × 10^−2^	7.81 × 10^−4^	1.10 × 10^−4^
Sodium citrate dihydrate	5.26 × 10^−2^	1.11 × 10^−3^	1.39 × 10^−4^
Potassium chloride	5.70 × 10^−2^	1.74 × 10^−3^	3.58 × 10^−4^
Lithium chloride	6.93 × 10^−2^	1.18 × 10^−3^	1.02 × 10^−4^
Benzylic alcohol	1.46 × 10^−1^	2.35 × 10^−3^	1.61 × 10^−4^
Sodium phosphate	1.50 × 10^−1^	2.19 × 10^−3^	4.42 × 10^−4^
Boric acid	1.70 × 10^−1^	2.47 × 10^−3^	1.80 × 10^−4^
Disodium EDTA	4.19 × 10^−1^	6.53 × 10^−3^	6.14 × 10^−4^
Dibasic sodium phosphate	5.87 × 10^−1^	6.15 × 10^−3^	6.52 × 10^−4^
Benzalkonium chloride	2.03	2.84 × 10^+7^	1.20 × 10^+7^

**Table 5 sensors-21-05879-t005:** Free energies of binding of different positive controls onto CB[6].

Ligand	Free Energy of Binding (kcal/mol)
Ephedrine	−3.0
Isoprenaline	−3.6
Lidocaine	−2.8
Octopamine	−2.9
Prilocaine	−3.6
Synephrine	−3.3

## Data Availability

Not applicable.

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
