# Peer review of "Supramolecular Atropine Potentiometric Sensor"

_sensors, 2021, doi:10.3390/s21175879_

Round 1
Reviewer 1 Report
We read with interest the article “Supramolecular atropine potentiometric sensor” by Ferreira et al. In this article, the researchers designed and tested a novel potentiometric PVC membrane sensor based on the formation of the ion association complex of the atropinum cation with cucur- 12bituril (CB). The aim of the researchers is to be able to determine atropine concentration and eventually use such a sensor as a quality control tool for hospital drugs containing atropine.
The sensor showed a fast, stable, near Nernstian response with a cationic slope of 58.7+0.6 mV/decade. Also, selectivity coefficients of atropine were investigated and showed for most studied cations and/or anions negligible interference which comes to support the robustness of the method. Moreover, the researchers endorsed their findings by the in-silico docking results they obtained which were furthermore supported by the results from the spectroscopic studies (both IR and H spectrometry) which revealed the existence of a guest-host interaction between atropine and CB6. Also, the direct determination of atropine which revealed good/significant results comes to support the usefulness of such a method and its promising applicability in the hospital field.
Comments:
I do believe that this article is suitable for and worthy of publication if some modifications are made to it; here are some of the points that I do believe can be worked on:
- The abstract directly describes the work that the researchers achieved without introducing the importance and relevance of the to-be-found results
- The materials and methods that the researchers used do align with the scientific question/hypothesis that is being asked
- However, a more profound search in the literature would definitely increase the reliability factor of this paper especially in the introduction part
- Some nuances in the used vocabulary of the introduction would be encouraged: i.e. use “because of” instead of “due” in line 37 as there is a pejorative connotation to the word “due”
- In the introduction, it would have been beneficial to introduce slightly the concept of PVC electrodes as ionic sensors; also, ending the introduction with assertive/conclusive claims right after stating the hypothesis is not recommended
- When it comes to the units, the editing should be reviewed: mS.cm-1 (not mS cm-1; line 82), mol.L-1 (not mol L-1; lines 84-95) and so on
- For equation 1, the researchers should have mentioned right under it what each abbreviation (Za, Zi, Ea and Ei mean): Za and Zi are the charges of atropine and interfering species ...
- Typo error in line 133: ph=6
- For the docking studies, the researchers did not explain why they chose those controls specifically (ephedrine, isoprenaline, octopamine …)
- The researchers did not explain how they calculated the LLLR and PDL
- Also, when it comes to checking the effect of pH on the electrodes’ potential, they only evaluated it for two fixed concentrations which are quite close to one another (maybe should have also done it for 1.00*10^-3 mol.L-1 and 1.00*10-6 mol.L-1)
- Moreover, when choosing which membrane to stick to for the rest of the study, they did chose membrane II with the slope of 60.31 mV.dec-1 but with an n=12 rather than any of the other membranes whose slopes were calculated with n=6; it would have been better to stick to the same n for all membrane types and then chose depending on the results obtained
- When discussing the guest-host interactions, the researchers should have also added some literature on such interactions with other ion-pairing agents other than CB such as phosphotungsate, valinomycin …
- For figure 1, designing a more complete representation of the crystallographic structure of the docking of atropine to CB6 that would also show van der Waals and dipole-dipole interactions in addition to the polar interactions would be more supportive in that the researchers mentioned all those interactions in-text
- There are some typos and editing errors that should be fixed specially: line 264 (coma should be at the end of a line and not at the beginning of one), line 266 (both peak intensity and shape), line 272 (to previously described: space to be added), line 273 (figure 2B - grey spectrum)
- They did state that CB6 in DMSO-d6 is much more soluble in the presence of atropine than alone but did not state the importance/relevance of such finding
- Maybe it would be better to rephrase the sentence (line 285) in a less conclusive manner i.e. instead of “allowing to justify” maybe say “which comes to support”
- Even though the researchers did try to verify the feasibility of the developed method by measuring, in four replicates, the atropine concentrations of certified samples, determination of atropine should have covered a greater concentration range (i.e. 10^-3 to 10^-6).
- Also, testing the efficacy of the studied atropine-based sensor on actual atropine-containing drugs used in hospitals such as atropine injection and eye drops would have come to support the researchers' claim in that their proposed sensor can be used as a quality control tool for hospital drugs.
- The authors’ conclusions might seem exaggerated for some in that a big part of the relevant and significant findings are based on computer-designed programs and are closer to the theoretical than the practical aspect
- A lot of the cited literature is actually more than a decade old; supporting such resources with newer/updated ones is definitely advisable
Reviewer 2 Report
Dear Sir,
The manuscript is very interesting since it presents a new technique for assesing atropine in various types of solutions. Since easy detection of this compound is important also from a forensic point of views, I suggest that in the Introduction paragraph further comments to be made, related to that topic. In the same time, some discussion should be made in the results paragraph concerning the LODs of the authors’ method compared to other techniques (for example, Brown et al. managed a detection limit of 0.75 micromoles using an electrochemiluminescent (ECL)-based sensors - Anal. Chem., 2019, 91(19), 12369-12376).
In the same time, it wold have been worth mentioning that complexes of CB with atropine have alredy been study, although it was with CB[7] (Karasova et al., Int J Mol Sci. 2020, 21(21), 7883, doi: 10.3390/ijms21217883).
The manuscript clearly needs a little polishing for the English language (for example there are numerous expressions such as “It was observed a negative correlation between the potential and the pH.” – “A negative correlation between the potential and the pH was observed.”)
The results are sound and the method is correct.
There are however some issues, as follows:
- a) the interaction between atropine and CB[6]: it was indeed demonstrated that the main interaction is between the nitrogen of the tropane ring and the portals carbonyl moieties. I agree that further hydrogen bonding can occur between the atropine’s -OH group and the same carbonyls in CB[6]. In the same time, an interaction between the mesomeric forms of the ester moiety in atropine and any carbonyl group in CB[6] is possible. However, I doubt of the existence of the interactions between the benzene ring and the carbonyl moiety (the cited paper is dealing with intramolecular n-pi interaction – here the authors assume intermolecular interactions; in the late 1990s there were some reports of interactions between aromatic compounds and polyacrylate derivatives on silica, but the nature of the interactions was not explained).
- b) In the IR spectra, the blue spectra that the authors atributed to atropine doesn’t looks like atropine, but more like atropine sulphate. CB[6] spectra seems all right, but the interpretation of the complex atropine-CB[6] should be correlated with the assumption made for the latter. Clear proofs of the inclusion of atropine in CB[6] are the presence in the IR spectra of the complexe of the 1600 cm bands of the benzene ring and the 1100 cm-1 adsorption band belonging to the C-N-C moiety of the tropane ring. In the same time, the region 2800-2900 shows a more diversified types of C-H bonds. Moreover, there is also the 3300 cm-1 band for the -OH of atropine.
- c) the NMR spectra are unconclusive: the atropine spectra is OK, but CB[6] is really bad and the NMR for the complex is missing. So either the authors come with a better 1H-NMR set of spectra, either they avoid this discussion.
Minor comment:
- in Fig. 1, both structures are on the same scale? To me, atropine seems represented at a slightly larger scale than CB[6].
I consider thus that the manuscript could be considered for publication after minor complement of information and a re-editing of the text for English language corrections (as well as some typos, e.g. “bands considering both i peak intensity and shape”).
Reviewer 3 Report
The manuscript deals with potentiometric sensors for atropine based on a molecule-selective membrane incorporating cucurbit [6]uril as a recognition element. In addition to quality checking, the detection and quantification of drugs potentially released in the environment are important and timely topics in line with Sensors. English writing should be improved, in particular in the result section that could be thoroughly spell-checked, although the text is still understandable. In addition, please address the following comments:
Please include a reference for the preparation protocol of the complex for IR analysis by mixing solid powders, and explain why this approach is more efficient than dissolving both powders, mixing the solutions and drying or precipitating the resulting complex. Moreover the formation of the complex may be strongly influenced by the presence of solvent molecules.
A great application of such sensor would lie in real time monitoring of e.g. environmental samples or body fluids, but the selectivity tests are only carried out in the presence of excipients. Could the authors add further details regarding the domain of application of the sensor? Is this focused solely on quality checks of final drug formulations?
The quality of Fig. 2 should be improved.
In 3.3, concentration units are missing.
In conclusion, the authors state “good selectivity, down to micromolar level, much better than other sensor reported in the literature”. The authors should more explicitly add values and references from the state of the art studies to prove their point and compare the performance of their sensor with that of others.
More details have to be given regarding the repeatability of electrode production, measurements and regeneration - how many electrodes were prepared and measurements carried out for each point? Is the detection reversible, have the electrodes been exposed to cycles of high/low concentrations of the analyte, or are the electrodes for single use? This information should be clearly stated upfront so that readers can appreciate the domain of applicability of the sensor.
